# Activity of Anti-Microbial Peptides (AMPs) against *Leishmania* and Other Parasites: An Overview

**DOI:** 10.3390/biom11070984

**Published:** 2021-07-04

**Authors:** Rima El-Dirany, Hawraa Shahrour, Zeinab Dirany, Fadi Abdel-Sater, Gustavo Gonzalez-Gaitano, Klaus Brandenburg, Guillermo Martinez de Tejada, Paul A. Nguewa

**Affiliations:** 1ISTUN Instituto de Salud Tropical, Department of Microbiology and Parasitology, IdiSNA (Navarra Institute for Health Research), University of Navarra, c/Irunlarrea 1, 31008 Pamplona, Navarra, Spain; reldirany@alumni.unav.es; 2Faculty of Sciences I, Lebanese University, Hadath 1003, Lebanon; hshahrour@alumni.unav.es (H.S.); fabdelsa@ul.edu.lb (F.A.-S.); 3Department of Microbiology and Parasitology, IdiSNA (Navarra Institute for Health Research), University of Navarra, 31008 Pamplona, Navarra, Spain; gmartinez@unav.es; 4Department of Chemistry, Faculty of Sciences, University of Navarra, 31080 Pamplona, Navarra, Spain; zdiranyahma@alumni.unav.es (Z.D.); gaitano@unav.es (G.G.-G.); 5Brandenburg Antiinfektiva GmbH, c/o Forschungszentrum Borstel, Leibniz Lungenzentrum, 23845 Borstel, Germany; kbranden@gmx.de

**Keywords:** anti-microbial peptides (AMPs), anti-bacterial, anti-fungal, anti-viral, anti-parasitic, anti-tumor, *Leishmania*, parasite, bacteria, Cathelicidin, Cecropin, Defensin, Dermaseptin, Eumentin, Histatin, Magainin, Melittin, Temporin

## Abstract

Anti-microbial peptides (AMPs), small biologically active molecules, produced by different organisms through their innate immune system, have become a considerable subject of interest in the request of novel therapeutics. Most of these peptides are cationic-amphipathic, exhibiting two main mechanisms of action, direct lysis and by modulating the immunity. The most commonly reported activity of AMPs is their anti-bacterial effects, although other effects, such as anti-fungal, anti-viral, and anti-parasitic, as well as anti-tumor mechanisms of action have also been described. Their anti-parasitic effect against leishmaniasis has been studied. Leishmaniasis is a neglected tropical disease. Currently among parasitic diseases, it is the second most threating illness after malaria. Clinical treatments, mainly antimonial derivatives, are related to drug resistance and some undesirable effects. Therefore, the development of new therapeutic agents has become a priority, and AMPs constitute a promising alternative. In this work, we describe the principal families of AMPs (melittin, cecropin, cathelicidin, defensin, magainin, temporin, dermaseptin, eumenitin, and histatin) exhibiting a potential anti-leishmanial activity, as well as their effectiveness against other microorganisms.

## 1. Introduction

Anti-microbial peptides (AMPs), also called host defense peptides (HDP), are a growing class of peptide-based molecules, with a wide spectrum of biological activities. They are small peptides, consisting of 5–100 amino acid residues with diverse molecular weights [1,2]. Several living organisms, such as bacteria, fungi, plants, invertebrates, non-mammalian vertebrates, and mammals, generate AMPs [3]. They are involved in innate immunity and the induction of resistance against such anti-microbial peptides is uncommon [4]. AMPs exhibit broad-spectrum anti-microbial properties, acting by the direct elimination of infectious pathogens (bacteria, viruses, fungi, and parasites) or by modulating the immune response. They activate and recruit immune cells resulting in the enhancement of the pathogen elimination and/or in the control of inflammation, wound healing, and angiogenesis [5,6].

Furthermore, AMPs may act as signaling molecules, biomarkers, and also as anti-tumor agents [7]. These peptides are mainly cationic and amphipathic, harboring hydrophobic residues (normally 50%). In total, more than 3000 natural AMPs have been identified so far (APD: Anti-microbial Peptide Database; http://aps.unmc.edu/AP/ (accessed on September 2020) and were principally from eukaryotes [2,8]. AMPs are not only isolated from amphibians, birds, fishes, mammals, insects and other invertebrates, and plants, but also from bacteria [3,9,10,11]. In insects, AMPs are produced in the fat body and the hemocytes or epithelia [12]. In vertebrates, AMPs are detected in mammals (in lymphocytes and leukocytes) [13,14]; they were also present in amphibian skin secretions [15] and epithelia [16]. Indeed, more than 300 distinct AMPs were isolated from the frog skin [17]. AMPs are also produced by human cells, including immune cells (phagocytes as well as lymphocytes) [18], and gastrointestinal epithelial cells [19].

AMPs can be divided into subgroups depending on their sequences and structures [20]. Several classes of AMPs have been identified, including cecropins, magainins, melittin, dermaseptins, defensins, and cathelicidins, which are cationic peptides of ~20–50 amino acids. Interestingly, cathelicidins and defensins are the main groups [21] and AMPs can be fully synthetized or modified chemically [22]. Such chemical modifications may allow altering of the target sites of the peptides and to enhance their resistance against proteolytic enzymes [23].

AMPs were first used to fight the antibiotic resistance of microorganisms [7], since these compounds are not affected by the mechanisms of bacterial resistance to conventional anti-microbials. The increase of multi-drug resistant (MDR) bacteria, due to the excessive use and misuse of antibiotics, is a critical issue. In the clinic, the death rates and the period of hospitalization is much higher in patients infected with drug-resistant organism [7,24], and the healthcare burden due to the proliferation of MDR bacteria has dramatically augmented in recent years [25]. This rapid increase in antibiotic resistance is alarming, according to the World Health Organization (WHO) [26]. In this sense, AMPs were highlighted as substitutes of conventional antibiotic agents, since these compounds can be readily modified to minimize their intrinsic limitations, such as toxicity, excessive length, and degradation by host proteases [27,28,29].

A common characteristic of AMPs is their fast inhibitory activity. In addition, some of them are able to even lysate within seconds following the interaction with the target’s membrane (Figure 1). This activity is mainly a result of the amphipathic conformation of AMPs that enhances their ability to contact the hydrophobic part of the lipid components and to the hydropholic part of phospholipid polar groups, causing the complete disruption and permeabilization of the bacterial membrane [1]. A wide range of their biological activities has been described, with the anti-microbial activity being the most commonly investigated, specifically against bacteria (anti-bacterial activity). When dealing with their activity on parasites, among the 990 active AMPs registered at the APD database, only 83 peptides have been assessed as anti-parasitic agents, mainly against malaria [30], while the anti-leishmanial activity of many AMPs families has only been studied in recent years.

Leishmaniasis, a parasitic illness caused by *Leishmania* species, is transmitted to humans through vectors (phlebotomine sand flies) [31]. Leishmaniasis is considered by WHO as a neglected tropical disease. After malaria, it is the most worrying parasitic illness worldwide [32]. About 0.7−1 million new cases of leishmaniasis are reported annually, generating around 20,000−30,000 deaths, according to WHO [33]. Leishmania parasites have a complex life cycle consisting of two stages: promastigote in the vector sandfly and amastigote in mammalian hosts. Promastigotes have a motile flagellum and elongated shape. Once infecting the macrophages, they differentiate to nonmotile amastigotes with very short flagellum and ovoid cell body [34]. Over 20 *Leishmania* species are known to be infective to humans, classified into old world (Mediterranean countries, Asia, and Africa) and new world parasites (America) [35]. On the other hand, among 800 species of sandflies, only *Phlebotomus* species and subspecies in the Old World and *Lutzomyia* in the New World, are proven as vectors for human leishmaniasis [36]. However, the parasite could be in few cases transmitted from mother to child during pregnancy [37], by syringe sharing [38], or blood transfusions [39]. Depending on the characteristics of the parasite and the vector sandfly, the disease is manifesting in three main types: cutaneous, mucocutaneus, and visceral. Cutaneous leishmaniasis constitutes 90% of the cases and causes skin lesions that can evolve to life-long ulcers while mucocutaneus leishmaniasis is a rare form of the disease associated with the inadequate treatment of the primary infection and causes both skin and mucosal ulcers. Visceral leishmaniasis occurs when the parasite reaches the internal organs, it is considered the most severe form and can be lethal if left untreated [40]. Generic pentavalent antimonials have been the first-line drugs against this pathology [41], but these present limitations, such as drug resistance and severe side effects [42]. Therefore, alternative treatments based on amphotericin B, miltefosine, and paramomycin, were also approved.

Amphotericin B is a polyene antibiotic that offers a high efficacy, but its high cost and difficulty of intravenous administration limit its clinical use [35,43]. Similarly, miltefosine, a recognized oral agent used for leishmaniasis treatment, is a costlier option compared to antimonials [44]. Paromomycin, a natural aminoglycoside antibiotic synthesized by *Streptomyces riomosus*, has also shown activity for the treatment of this disease. Whereas, these treatments are limited by several side effects, ranging from the mild pain at the injection site to the development of hepatic and renal toxicity [45].

Due to the aforementioned limitations that reduce the efficiency of all these compounds, in addition to the lack of an effective human vaccine against leishmaniasis [46], new therapeutic strategies and innovative reformulations are urgently needed. In this regard, isoselenocyanate derivatives, which are well known for their anti-tumor activity, were demonstrated for the first time to have potential effect on leishmaniasis [46]. Furthermore, since selenium derivatives are able to reduce parasitemia [47], some selenocompounds were also investigated and reported to exhibit leishmanicidal effect [48]. More recently, a new reformulation of miltefosine has been developed, based in the incorporation of this compound to poly(ethylene)oxide (PEO)-based polymeric micelles [49]. These formulations improved the drug’s effectiveness against amastigotes, revealing itself as an encouraging methodology for the cure of leishmaniasis.

A different therapeutic approach involves the use of AMPs. Numerous AMPs, like melittin, cecropin, cathelicidin, defensin, magainin, temporin, dermaseptin, eumenitin, and histatin, have proven to be active on different *Leishmania* species (Table 1).

The general mechanisms of action of these AMPs against microorganisms are summarized in Figure 1.

In this review, we mainly describe the principal families of AMPs exhibiting a potential anti-leishmanial activity, as well as in their effectiveness against other parasites and bacteria.

## 2. Families of AMPs and Their Anti-Infectious Activities

### 2.1. Melittin

#### 2.1.1. Melittin against Leishmaniasis

Melittin is a potent cationic peptide found in the bee venom of *Apis mellifera* (European honey bee), and is used as an anti-inflammatory agent [75,76]. It is considered as the main bioactive constituent of the bee venom [77]. Melittin is a linear peptide consisting of 26 amino acid residues (H-Gly-Ile-Gly-Ala-Val-Leu-Lys-Val-Leu-Thr-Thr-Gly-Leu-Pro-Ala-Leu-Ile-Ser-Trp-Ile-Lys-Arg-Lys-Arg-Gln-Gln-OH) (Table 1) and harbors amphiphilic properties [76] which enhance its ability to induce membrane permeabilization in both prokaryotic and eukaryotic cells [78]. Therefore, melittin displays several biological activities. Besides its haemolytic activity, melittin can also be used as anti-microbial [79], anti-viral, anti-fungal [75] and anti-tumoral agent [77]. In addition, inhibitory effects of this peptide on parasitic protozoa have been described without significant toxicity [80].

Several studies have reported the effectiveness of melittin against leishmaniasis. It was demonstrated that melittin is active on *L. donovani* promastigotes, showing an IC_50_ value (50% inhibitory concentration) lower than 1.5 µM [81]. Melittin was purchased from Sigma (St. Louis, MO, USA) [81]. Another study has revealed that melittin was effective on *L. major* and *L. panamensis promastigotes*, with IC_50_ values of 74.01 µg/mL and 100 µg/mL, respectively [79]. Lastly, Pereira et al. revealed the potential of melittin against *L. infantum* in both promastigotes and amastigotes forms [75].

#### 2.1.2. Activity on Other Parasites

Melittin has been proved to act against *Trypanosoma cruzi* parasite, responsible for Chagas disease. This AMP showed a lytic activity on different developmental forms of the parasite: the epimastigote (vector stage), the trypomastigote (infective non proliferative stage) and the intracellular amastigote (proliferative stage) [80]. Interestingly, melittin affected the viability of *T. cruzi* amastigotes at lower concentrations compared to the concentrations inducing toxicity in mammalian cells (100-fold lower) [80]. Melittin was obtained from Sigma (St. Louis, MO, USA) [80]. Furthermore, the peptide caused morphological alterations in the parasite on nuclei, mitochondria, and membrane extensions. The mechanisms of death activated by such an AMP might be apoptosis as well as autophagy [80] (Figure 1).

Moreover, new studies demonstrated that the hybrid peptide CM11, composed of melittin and another AMP called cecropin, exerts a potent killing effect against *Entamoeba histolytica* intestinal parasite.

The cytotoxicity of CM11 was evaluated on *E. histolytica* alone, and co-cultured with the human colonic carcinoma cell line (Caco-2). CM11 peptide exhibited an anti-parasitic activity of 93.7% on *E. histolytica* trophozoites alone, while 63.5% of trophozoites were killed in the co-culture at the same peptide concentration (24 µg/mL). This result suggested that the co-culture of the parasite with the host epithelial cells conferred it a higher resistance to the peptide [82]. Melittin was active against sporogenic stages of *Plasmodium*: in vitro assays showed that, at 50 µM, the peptide was able to kill 100% of *P. berghei* ookinetes following 24 h treatment. In vivo effect of melittin on mosquitos infected with *Plasmodium* parasites was also studied. *Anopheles stephensi* mosquitoes were fed on blood containing rodent malaria *P. berghei* gametocytes, while in *An. gambiae* mosquitoes, the feeding blood contained human malaria *P. falciparum* gametocytes both supplemented with 50 µM of melittin. The compound reduced the infection and the parasite burden inside the host cells by 10% and 86%, respectively, after an infection by *P. berghei*, and by 60% and 57%, respectively, when cells were parasited by *P. falciparum*. Interestingly, when added to a non-infective blood meal, melittin showed no impact on mosquito longevity and fecundity [83].

### 2.2. Cecropin

#### 2.2.1. Cecropin and Leishmania

Cecropins anti-microbial peptides, initially found in the hemolymph of Hyalophora cecropia, are a main part of the innate immunity of insects [84]. They are also produced naturally by other kinds of insects (*Galleria mellonella*, tsetse flies, Drosophila) after bacterial infections [85,86]. They are linear cationic peptides having a sequence of 31–37 amino acids that differ greatly depending on the species [84]. Cecropins are considered an essential family of amphipathic proteins in insects. Cecropin homologues have been detected in mammals as well, showing the high level of conservation of the non-specific (i.e., innate) immune system between insects and mammals.

Cecropin family consists of five subtypes (A−E) in addition to other cecropin-like peptides, such as spodopsins papiliocins, enbocins, stomoxins, and sarcotoxins [87,88,89,90,91]. Regardless their potential bactericidal activity [92], cecropins have shown a promising activity against tumor cell proliferation [93], in addition to their significant anti-fungal activity [94]. Likewise, the activity of cecropins against *Leishmania* parasites has been explored in several studies. Cecropin A can be purchased from Peninsula Laboratories (Belmont, CA, USA) [81]. Cecropin-D (Table 1), which is Cecropin-A homologous, has shown reduction of the parasite growth of *Leishmania (V) panamensis* promastigotes by 57% when the concentration was 100 µM [86]. In another study, Cecropin-A extracted from *Hyalophora* and *Drosophila* (Table 1) was demonstrated to inhibit the *Leishmania aethiopica* growth (amastigotes and promastigotes)*,* with lower concentrations against amastigotes (50% inhibition at concentrations ~0.250 mg/mL). Interestingly, those peptides showed no hemolytic activity at the same concentrations [85]. Another study reported that cecropin-A was the most active on *L. panamensis* amastigotes, among three tested cecropins (cecropin-A, -B, -C) isolated from the giant silkworm *Hyalophora cecropia* hemolymph. Interestingly, Cecropin-A was neither cytotoxic nor haemolytic at the corresponding concentrations [79].

#### 2.2.2. Activity on Other Parasites

Several studies reported the anti-parasitic activity of cecropins. Cecropin-B was active against a variety of *Plasmodium* species when administered to its mosquito vector. When the peptide at 0.5 µg/µL of was injected into anopheline mosquitos five or more days after a *Plasmodium*-infected blood meal, the effect of cecropin on the development of oocysts was dramatic. Light microscopy imaging analysis showed that the peptide also induced deformation of the developing oocysts [95]. In another study, the *Plasmodium* vector *Anopheles gambiae*, was genetically modified to express the anti-microbial peptide cecropin-A. The number of oocysts was reduced by 60% in genetically-modified *A. gambiae* mosquitoes compared to non-transgenic ones, both infected with *Plasmodium berghei* [96]. On the other hand, the infection of the tsetse fly *Glossina morsitans* with *Trypanosoma brucei brucei* induced the synthesis of cecropin, which represented a marker of the humoral immunity [97]. Lastly, SB-37, a derivate of cecropin-B, demonstrated a lethal effect on *Plasmodium falciparum* and *Trypanosoma cruzi* parasites without cytotoxicity in host eukaryotic cells (erythrocytes and Vero cell line) [98].

### 2.3. Cathelicidin

#### 2.3.1. Cathelicidin and Leishmania

Cathelicidins constitute another well-characterized AMP family with members distributed in different mammal species, including pig, cow, rabbit, and humans. Cathelicidins are cationic, amphiphilic peptides having 12–97 amino acids in length [99]. There is a wide variety of cathelicidin-derived peptides which differ in structure and activity (Table 1). Cathelicidin derivatives are generated through the proteolytic cleavage of the cathelin-domain, which is conserved in all the family members. This enzymatic process allows the mature COOH-terminal anti-microbial peptide to be released [100].

Regarding properties other than anti-microbial, some members of this family were reported to possess host defense capacity, and described to induce wound healing by activating mesenchymal cells [101]. Many other activities of cathelicidin members have been observed as well. CAMP, the sole cathelicidin-type peptide identified so far in humans, is mainly detected in the cells involved in the host defense response (neutrophils, macrophages), endothelial and epithelial cells. In addition to its direct role in fighting microorganisms, CAMP could also act indirectly by regulating apoptosis, angiogenesis, cell proliferation, inflammatory reactions, cytokine release, and cell cycle arrest (Figure 1). Recently, CAMP has been described to act as anti-oncogenic agent in breast cancer [101]. Regarding cutaneous leishmaniasis (CL), a study in Ethiopian patients proved the existence of high expression of CAMP in skin lesions [102].

Moreover, recombinant cathelicidin was reported to be *leishmanicidal* [103]. Chathelicidin LL-37 was produced by Thermo-Fischer [103]. In another study, LL-37 peptide (Table 1)**,** which is a cathelicidin-derived peptide obtained by the cleavage of the human cationic anti-microbial peptide-18 (hCAP-18) encoded by CAMP, demonstrated an effectiveness against leishmaniasis: LL-37 was able to decrease around 50% of the viability of the promastigotes of *L. donovani* compared to the untreated control. Furthermore, in their intramacrophage stage, *L. donovani* and *L. major* amastigotes were equally susceptible to LL-37 peptide [104]. Likewise, the bovine myeloid anti-microbial peptide (BMAP-28) was also studied against *Leishmaniasis*. BMAP-28 peptide is a cathelicidin consisting of 28 amino acids (Table 1) isolated from bovine neutrophils. RI-BMAP-28, L-BMAP-28, andD-BMAP-28 were considerably active in vitro on *Leishmania* promastigotes, and the D-isoform was the most effective in the reduction of promastigote viability. Furthermore, BMAP-28 peptides have shown activity against amastigotes. Hence, RI-BMAP-28, L-BMAP-28, and D-BMAP-28, could be promising alternative treatments of leishmaniasis [105].

#### 2.3.2. Activity on Other Parasites

The equine anti-microbial peptide eCATH1 was found to display a trypanocidal activity against three species of Trypanozoon parasites: *T. equiperdum, T. evansi,* and *T. brucei brucei*, responsible for animal trypanosomiasis dourine, surra, and nagana, respectively. In vitro studies showed that eCATH1 acts similarly against all trypanozoon parasites exhibiting an IC_50_ = 9.5 µM [106]. eCATH1 was able to modify the plasma membrane permeability inducing autophagic, necrotic cell death, or apoptosis. In addition, the disruption of the potential of the mitochondrial membrane was rapidly observed after 15 min of treatment with eCATH1 at its IC_50_ (=9.5 µM). Moreover, dramatic structural changes were reported, like membrane blebbing of organelles, cytoplasmic vacuolization, trypanosome body swelling, perturbation, and loss of microtubules of the membrane. Furthermore, the administration of eCATH1 at 10 mg/kg to *T. equiperdum*-infected animals delayed mouse death. In accordance with these findings, it was hypothesized that trypanosomes would be unlikely to develop resistance to eCATH1, because of the unique combination of mechanisms of action, differing from the classic membrane disruption, so common in other AMPs [106].

Within this family, LZ1, a well-studied peptide found in snakes, was shown to possess promising anti-plasmodial effects. The percentage of the asexual blood stage of *P. falciparum* parasites decreased in vitro by an average of 61% after treatment with a low concentration of LZ1. The in vivo anti-plasmodial effect of LZ1 was assessed in *P. berghei* infected mouse models, which displayed prolonged survival and decreased in the parasitemia rate compared to uninfected animals. Interestingly, LZ1 induced modulated the immunity of *P. berghei* infected mice by decreasing the overexpression of pro-inflammatory factors (IFN-γ, IL-6, TNF-α,); thus, it attenuated liver damage resulting from malarial infection. An additional mechanism of LZ1 was found to selectively lower the synthesis of ATP in infected RBC, by the inhibition of pyruvate kinase function [107].

### 2.4. Defensin

#### 2.4.1. Defensin and Leishmania

Defensins are part of the first groups of anti-microbial peptides identified in mammalian organisms. Like the other AMPs described so far, they are characterized by a conserved six-cysteine signature, and are classified into three sub-groups, α, β, and θ [108]. Defensins are derived from human beings and other organisms [109,110]. In humans, they are a key element in the innate immunity, playing a crucial role in host defense against microbial infections. In addition, defensins are an important component of acquired immunity, since they can induce the migration of different immune cells (mast cells, T-lymphocytes, dendritic cells, and monocytes) to the infection site while enhancing macrophage-mediated phagocytosis [111] (Figure 1). Notably, defensin-like peptides were also found in plants. Whereas, human defensins consist of 29–35 amino acids [109], their plant counterparts consist of 45–54 amino acids [112].

Defensins exhibit a wide variety of anti-microbial effects, such as anti-bacterial [109,113], anti-fungal [110,114,115], anti-viral [116,117], and anti-leishmanial activities. A recent study demonstrated that mouse beta defensins mBD1, mBD2, and mBD3 (Table 1) were upregulated in C57BL/6 mice, a mouse strain well-known for its resistance to *Leishmania* infection. Insensitivity of C57BL/6 mice to *Leishmania* may also be the consequence of additional events of immune system activation. In this respect, it was reported that CL by *L. major* parasites induced secretion of interleukin (IL)-12 by macrophages. In turn, these cells induced the expression of different cytokines, like INF- γ by T lymphocyte cells, leading to macrophage activation and the killing of the intramacrophage form of the parasite [111] (Figure 1).

Due to the important role of defensins in the resistance of plants to pathogen infections [118], plant defensins were classified as plant pathogenesis-related (PR) proteins within the PR-12 group [112]. *Vigna unguiculata* defensin (Vu-Def) (Table 1), a defensin of plant origin, was found to be effective against *L. amazonensis*. Testing of progressively shortened variants of Vu-Def allowed the identification of the key domain implicated in the anti-microbial activity of the peptide, designated as γ-core. This domain preserved the whole peptide biological activity and was identified as a conserved region formed by a few amino-acid residues. Notably, plant defensins have shown no toxicity on mammal cells [112].

#### 2.4.2. Activity on Other Parasites

Human defensin α-1 was reported to have trypanocidal effect on T. *cruzi*. The peptide displayed killing effects on amastigotes and trypomastigotes (at 3.7–35 µM). Human defensin α-1 acts by generating pores on the membrane of the trypomastigote, besides, it induces the fragmentation of their mitochondrial and nuclear DNA (Figure 1). Interestingly, the infectivity of trypomastigotes in human epithelial cell line (HeLa cells) was reduced after the pretreatment of the parasites with a sublethal dose of defensin α-1 anti-microbial peptide [18].

Notably, defensins from the European tick vector *Ixodes ricinus* exerted an anti-microbial activity against *P. falciparum* which was conserved through evolution. Briefly, the sequence of the common defensin ancestor shared by scorpions and ticks (so-called Defensin Ancestor STiDA), was synthesized and tested against *P. faliciparum*, and its anti-microbial activity was compared with that of extant ticks’ defensins. Interestingly, in vitro, STiDA significantly inhibited the parasite growth with a potency similar to that of extant tick defensins [119]. Similarly, *I. ricinus* defensins were reported to have anti-plasmodial activity against *P. chabaudi* in mice. Defensins considerably reduced the parasitemia 1 h and 12 h after their administration at a dose of 120 μL of 1 mg/mL solution [120].

Human β-defensin-2 (HBD2), from intestinal epithelial cells (IEC), was active against *Toxoplasma gondii* (type I, II, and III). Pretreatments of parasites with synthetic HBD2 at 25 µM and 50 µM concentrations significantly decreased their infectivity.

Nevertheless, the high-virulence *T. gondii* (type I) repressed the early expression of HBD2 gene in IEC, while the low-virulence strains (type II and III) strongly stimulated it. These outcomes proved the role of human β-defensin-2 as an anti-microbial agent in innate immune response against *T. gondii* [121].

A further study proved the parasiticidal effect of human β-defensin-1 and -2 against *Cryptosporidium parvum* parasites by decreasing their infectivity and viability. These AMPs lead to the disruption of the membrane, reducing the osmoregulation and, finally, inducing cell death [122].

### 2.5. Magainin

#### 2.5.1. Magainin and Leishmania

Another well-studied family of α-helical peptides with similar mechanism of action to that of melittin are the magainins [123]. Magainin is a 23-residue peptide synthesized by the African clawed frog (*Xenopus laevis*) [124]. As other AMPs, magainin peptides exert their activity through insertion into cell membranes after interacting with negatively charged phospholipids and finally leading to cell lysis [125,126] (Figure 1). Magainins present low toxicity towards red blood cell membranes [127] and these peptides showed activity against microbes and were effective as anti-tumor agents [128,129].

Different magainins have been reported to be active on *Leishmania* protozoan. Among them, two hydrophobic magainin-2 analogues, MG-H1 and MG-H2, and their parental peptide, F5W-magainin-2 (Table 1), were tested on *L. donovani*. The results showed that the three peptides inhibited *L. donovani* promastigotes proliferation at micromolar concentrations, MG-H2 exhibiting the most potent activity [130]. Similar to other leishmanicidal membrane-disrupting peptides (cecropin-A-melittin hybrids and dermaseptins), the mechanism of action of the aforementioned magainins (magainin-2 analogues, MG-H1 and MG-H2) was dependent on parasite membrane disruption followed by induction of a fast bioenergetics collapse [130] (Figure 1). More recently, pexiganan peptide (Table 1), a synthetic magainin analog rich in lysine, has revealed apoptotic effect in *Leishmania* promastigotes [131] (Figure 1).

#### 2.5.2. Activity on Other Parasites

Magainin-2, a vertebrate polycationic peptide, exhibits cytotoxic effects against *Cryptosporidium parvum* sporozoites. After 20- and 60-min exposure to the AMP, at 100 and 10 µg/mL, respectively, the percentage of sporozoites viability decreased significantly and reached 9.7%. In contrast, on *C. parvum* oocysts, magainin-2 did not completely reduce the oocyst growth, and the percentage of viability remained above 65% after 180 min of exposure to a high peptide concentration (100 µg/mL). Concerning the mode of action of magainin-2, it has been hypothesized that the molecule could alter the apical complex of the sporozoite containing the ligands involved in the attachment and invasion of the host epithelial cells. In contrast, inside the oocysts, sporozoites are protected by a thick wall, which may explain the low effect of the peptide against non-excysted organisms [132].

### 2.6. Temporin

#### 2.6.1. Temporin and Leishmania

Temporins are natural host defense AMPs isolated from frog’s skin and consist of 10 to 17 amino acids [133]. They belong to α-helical AMPs with highly cationic and amphipathic properties that allow them to target different pathogens, including bacteria [134,135], viruses [136], filamentous fungi [137,138], and parasites [139]. These same properties are also responsible for their significant hemolytic activity and cytotoxicity [140]. So far, 130 peptides of this family have been identified. Temporin-A (TA), temporin-B (TB), and temporin-L (TL) have been largely studied since they are highly active against several microorganisms.

Both TA and TB (Table 1) were reported to display activity on *L. donovani* promastigotes and *L. pifanoi* amastigotes [141]. Their leishmanicidal action is favored by their capacity to induce membrane permeation causing severe damage to the parasite membrane [141] (Figure 1). Likewise, temporin SHd (Table 1) showed effective activity against promastigote forms of numerous *Leishmania* (*L. infantum*, *L. major, L. tropica, L. amazonensis*, and *L. braziliensis*) with similar mechanism of action. SHd was also active against *L. infatum* axenic amastigotes [141].

Recently, the anti-microbial activity of the peptide temporin-SHe (Table 1), a temporin-SH paralog from the Sahara frog (*Pelophylax saharicus*) was investigated [142]. Temporin-SHe was active on *L. braziliensis* and *L. major* promastigotes at 10.5 and 11.6 µM, respectively. It was highly potent against *L. infantum* as well, at lower IC_50_ value (4.6 µM).

#### 2.6.2. Activity on Other Parasites

Temporizin (an artificial hybrid peptide containing the N-terminal region of temporin A) and Temporizin-1 (a modification of Temporizin) showed promising activity against *Trypanosoma cruzi*. Flow cytometry assay revealed that temporizin-1 eliminated 57% of the parasites while temporizin killed 65%. The EC_50_ values obtained by the MTT assay were 887 ng/mL for temporizin-1 and 849 ng/mL for temporizin. Interestingly, both peptides induced intracellular alterations like chromatin condensation and mitochondrial cristae disorder (Figure 1). However, temporizin and temporizin-1 seemed not to affect *T. cruzi* cytoplasmic membrane, suggesting their ability to modify the fluidity of trypanosome cytoplasmic membrane [143].

Among temporins-SH isolated from the North African ranid frog *Pelophylax saharicus*, SHa has emerged as a potent AMP. SHa and its analog [K^3^]SHa were tested on *T. brucei gambiense* and *T. cruzi*. SHa and [K^3^]SHa exerted trypanocidal effect at low concentrations (IC_50_ ~ 10–17 μM). Morphological changes were observed in *T. cruzi* epimastigotes treated with 5 μM [K3]SHa during half an hour. The peptide damaged the cell body and the flagellum, modifying cell morphology, and indicating that temporin acts through a membranolytic mechanism [139] (Figure 1).

### 2.7. Dermaseptin

#### 2.7.1. Dermaseptin against Leishmaniasis

Dermaseptins are natural polycationic peptides secreted by the skin of amphibians as a defense strategy against microbes. They are typically constituted of 27–34 amino acids that greatly vary from one peptide to another. However, they all share a cationic amphipathic nature. Dermaseptins are lethal at very low doses against several microorganisms (bacteria, fungi, parasites, yeast, and enveloped viruses). Dermaseptins are not toxic to mammalian cells, except for dermaseptin S4 which displays potent hemolytic and anti-protozoan effects [144]. The first described member of the family, dermaseptin S1 (Table 1), was reported to have anti-leishmanial activity against *L. panamensis* [79]. Moreover, Dermaseptins S1–S5 was lethal against *L. mexicana* in its promastigote form at low concentrations [79]. They act by inducing cell membrane disruption leading to parasite death [145] (Figure 1).

Dermaseptin 01 (DS 01, Table 1), a synthetic dermaseptin peptide, was also reported to be active against *L. amazonensis* in the promastigote form [146,147].

More recently, encapsulation of dermaseptin S1 in Cry3Aa crystals, was shown to enhance its effectiveness on intracellular *Leishmania* parasites (*L. amazonensis* and *L. donovani*) [148]. Dermaseptin DS1 peptide was purchased from Pepmic Company (Suzhou, China) [148].

#### 2.7.2. Activity on Other Parasites

DS 01 was reported to be active against *Schistosoma mansoni* helminth, responsible for human schistosomiasis. At 100 µg/mL, DS 01 decreased the worm motility and killed all worms within 48 h. Furthermore, DS 01 has shown an effect on the reproductive fitness of adult worms. This deleterious effect is associated with shape changes on the tegument of *S. mansoni*, a critical organelle during the infection and survival in the host [149]. In another study, DS 01 was found to have anti-*Trypanosoma cruzi* effect. At 6 µM, this AMP reduced the protozoan cell population to a non-detectable level after 2 h of incubation. DS 01 exerted its trymanocidal activity by inducing the membrane disruption and cell leakage. On the other hand, DS 01 was not hemolytic against red blood cells, suggesting that it could be used in systemic therapy [150]. Phylloseptins, another family of dermaseptins discovered in the skin of *Phyllomedusa*, also showed anti-*Trypanosoma cruzi* activity. Among this family, PS-4 and PS-5 were highly effective on *T. cruzi* trypomastigotes (IC_50_ = 5.1 and 4.9 µM, respectively) [151].

### 2.8. Eumenitin and Leishmania

Eumenitin is a recently identified anti-microbial peptide, discovered in 2006 by Konno et al. [65]. These investigators isolated eumenitin from the venom of *Eumenes rubronotatus*. Eumenitin is composed of 15 amino-acids (LNLKGIFKKVASLLT) (Table 1) and is predicted to adopt a linear α-helical structure [65].

This AMP has shown activity against *L. major* promastigotes. Furthermore, a study carried out by Rangel et al. reported that both peptides, eumenitin-F (LNLKGLFKKVASLLT) isolated from *Eumenes fraterculus* and eumenitin-R (LNLKGLIKKVASLLN) isolated from E. rubrofemoratus (Table 1), displayed anti-leishmanial activity against promastigotes of *L. major* (Table 1) [152,153].

### 2.9. Histatin Effect on Leishmania

Histatins are human oral anti-microbial peptides secreted by the salivary glands into the saliva and related to immunity [154]. The histatin family contains 12 small histidine-rich cationic AMPs with the most abundant ones being histatin 1, 3, and 5 [155]. The other histatins are known to be proteolytic derivatives of histatins 1 and 3 [156]. Histatins are effective on several microbes. Regarding their anti-leishmaniasis activity, only Hst5, its D-enantiomer and its synthetic analog Dhvar4 (Table 1) have been studied on *L. donovani* promastigotes and *L. pifanoi* amastigotes. Hst5 was active on *Leishmania* at micromolar concentrations (lethal doses 50~7.3 µM on promastigotes and ~14.4 µM on amastigotes) [157]. D- Hst5 and Dhvar4 were more active on both parasite forms than Hst-5 [157].

## 3. Conclusions and Future Perspectives

As shown above, several lines of evidence show that AMPs are very promising anti-parasitic agents: (a) they are rapidly lethal against *Leishmania* and other parasites at doses comparable to those of current treatments; (b) they have a broad spectrum of bactericidal activity, a property that makes them very valuable for the treatment of polymicrobial infections; (c) the emergence of resistance to AMPs is unlikely, since these agents target cell structures that are essential for the organism and very conserved at the molecular level; (d) some of them have immunomodulating properties that enhance pathogen elimination; (e) their biological activity can be repeatedly improved by conducting studies of structure-activity relationship (SAR); and (f) a particularly promising field of research with AMPs involves the development of therapies based on combinations of AMPs with conventional anti-parasitic agents.

In spite of all these attractive features, AMPs also have limitations that may restrict their therapeutic use. First, the activity of many of these peptides is reduced or even abrogated in the presence of physiological concentrations of salts or other biological compounds. In addition, some AMPs are degraded by serum proteases, and this effect greatly decreases their in vivo half-life. Finally, due to their unspecific mechanism of action, these agents exhibit some level of cytotoxicity close to their therapeutic concentration [158]. Nevertheless, for the topical treatment of skin diseases (e.g., cutaneous leishmaniasis), these drawbacks should not hinder at all the development of AMP-based treatments in the near future.

Because of the limitations of the existing anti-parasitic treatments, the reported features of AMPs make them promising candidates to replace current available therapies. Some pharmaceutical companies are developing anti-bacterial and anti-fungal drugs based on natural peptides and it is very likely for such peptides to progress into clinical development. Specifically, in the case of cutaneous leishmaniasis, we expect AMP-based treatments to get approval for clinical use in very few years.

## Figures and Tables

**Figure 1 biomolecules-11-00984-f001:**
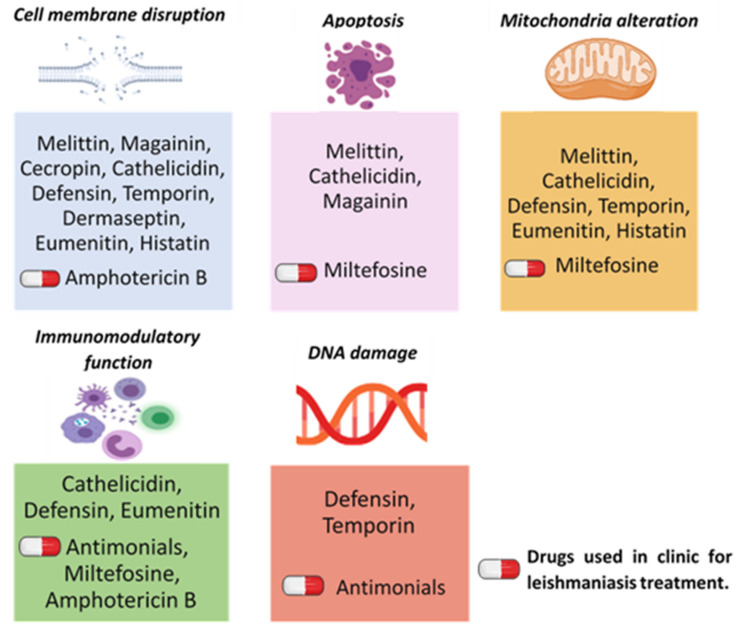
General mechanisms of action of the different families of AMPs with anti-leishmanial activity. **Cell membrane disruption** is the main mechanism of action adopted by all the described AMPs [50,51,52,53,54,55,56,57,58,59] While some of them present additional mechanisms of action (apoptosis, mitochondrial dysfunction, immune response modulation, and DNA damage) [60,61,62,63,64,65,66]. On the other hand, current drugs used in clinic for leishmaniasis treatment were found to exert similar mechanisms of action. For example, Amphotericin B principally affects the cell membrane [67] and can also modulate the immune response [68]. Miltefosine generates cell death mechanism (apoptosis) [69,70], affects the mitochondrial function [71] and the immune response [72]. Pentavalent antimonials cause DNA damage and can indirectly act by regulating the immune response [73,74].

**Table 1 biomolecules-11-00984-t001:** Some AMPs exhibiting leishmanicidal activity.

AMPs Family	Peptide Name	Sequence	*Leishmania* species	Reference
Melittin	Melittin	GIGAVLTTGLPALISWIKRKRQQ	*L. major*	(Pereira et al., 2016)
			*L. panamensis*	
			*L. donovani* promastigotes	(Pérez-Cordero et al., 2011)
			*L. infantum* promastigotes and amastigotes	(Díaz-Achirica et al., 1998)
Cecropin	Cecropin-A	KWKLFKKIEKVGQNIRDGIIKAGPAVAWVGQATQIAK	*Leishmania aethiopic*	(Pérez-Cordero et al., 2011)
			*L.panamensis* amastigotes	(Akuffo et al., 1998)
	Cecropin-D	ENFFKEIERAGQRIRDAIISAAPAVETLAQAQKIIKGGD	*Leishmania (V)*	(Patiño-Márquez et al., 2018)
			*L. panamensis* promastigotes	
Cathelicidin	LL-37	LLGDFFRKSKEKIGKEFKRIVQRIKDFLRNLVPRTES	*L. donovani* promastigotes and amastigotes	(Marr et al., 2016)
			*L. major* amastigotes	
	RI-BMAP-28	GIRIIPVIIPGYKKWARLIKRGLSRLGG	*L. major* promastigote	(Marr et al., 2016)
				(Lynn et al., 2011)
	D-BMAP-28	GGLRSLGRKILRAWKKYGPIIVPIIRIG	*L. major* promastigote	(Marr et al., 2016) (Lynn et al., 2011)
Defensin	MBD1	MKTHYFLLVMICFLFSQMEPGVGILTSLGRRTDQYKCLQ HGGFCLRSSCPSNTKLQGTCKPDKPNCCK	*L. major*	(Daneshvar et al., 2018)
	MBD2	MRTLCSLLLICCLLFSYTTPAVGSLKSIGYEAELDHCHTN GGYCVRAICPPSARRPGSCFPEKNPCCKYMK	*L. major*	(Daneshvar et al., 2018)
	MBD3	MRIHYLLFAFLLVLLSPPAAFSKKI--- NNPVSCLRKGGRCWNR-CIGNTRQIGSCGVPFLKCCKRK	*L. major*	(Daneshvar et al., 2018)
	Vu-Def	MKTCENLADTYRGP	*L. amazonensis*	(dos Santos et al., 2010)
Magainin	MG-H1	GIKKFLHIIWKFIKAFVGEIMNS	*L. donovani* promastigotes	(Guerrero et al., 2004)
	MG-H2	IIKKFLHSIWKFGKAFVGEIMNI	*L. donovani* promastigotes	(Guerrero et al., 2004)
	F5W-magainin 2	GIGKWLHSAKKFGKAFVGEIMNS	*L. donovani* promastigotes	(Guerrero et al., 2004)
	Pexiganan	GIGKFLKKAKKFGKAFVKILKK	*L. major*	(Zhang et al., 2015) (Kulkarni et al., 2009)
Temporin	Temporin A	FLPLIGRVLSGIL	*L. donovani* promastigotes	(Mangoni et al., 2005)
	Temporin B	LLPIVGNLLKSLL	*L. pifanoi* amastigotes	(Mangoni et al., 2005)
	Temporin-She	FLPALAGIAGLLGKIF	*L. braziliensis, L. major*	(André et al., 2020)
			*L. infantum*	
	SHd	FLPAALAGIGGILGKLF	*L. infantum*, *L. major, L. tropica, L. amazonensis*, and *L. braziliensis* promastigotes	(Mangoni et al., 2005)
			*L. infantum* axenic amastigotes	
Dermaseptin	DS 01	GLWSTIKQKGKEAAIAAAKAAGQAALGAL	*L. amazonensis* promastigotes	(Salay et al., 2011)
	Dermaseptin S1	ALWKTMLKKLGTMALHAGKAALGAAADTISQGTQ	*L. panamensis*	(Pérez-Cordero et al., 2011) (Yang et al., 2019)
			*L. mexicana* promastigotes	
			*L. donovani*	
			*L. amazonensis* amastigotes	
Eumenitin	Eumenitin	LNLKGIFKKVASLLT	*L. major* Promastigote	(Sabiá et al., 2019)
	Eumenitin-F	LNLKGLFKKVASLLT	*L. major* Promastigote	(Sabiá et al., 2019)
	Eumenitin R	LNLKGLIKKVASLLN	*L. major* Promastigote	(Sabiá et al., 2019)
Histatin	Hst5	DSHAKRHHGYKRKFHEKHHSHRGY	*L. donovani* promastigotes	(Luque-Ortega et al., 2008)
			*L. pifanoi* axenic	
	D- Hst5	D- DSHAKRHHGYKRKFHEKHHSHRGY	*L. donovani* promastigotes	(Luque-Ortega et al., 2008)
			*L. pifanoi* axenic	
	Dhvar4	KRLFKKLLFSLRKY	*L. donovani* promastigotes	(Luque-Ortega et al., 2008)
			*L. pifanoi* axenic	

## Data Availability

Not applicable.

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
