# Peer review of "Activity of Anti-Microbial Peptides (AMPs) against Leishmania and Other Parasites: An Overview"

_biomolecules, 2021, doi:10.3390/biom11070984_

Round 1

Reviewer 1 Report

The title of the paper: "Activity of antimicrobial peptides (AMPs) against Leishmania, other parasites and bacteria: an overview." is too much broad. Probably it should be focused only on the AMPs activity against parasites, including Leishmania. The initial part of the Introduction (lanes 45-58) need to be improved taking into consideration that AMPs have been isolated from a wide range of different sources (why amphibians and not fishes are cited, as an example). Leishmaniasis (lane 83) should be better introduced considering that it is one of the focus of the paper.  At the end of the paper a paragraph "Conclusions" should be added in which the authors should describe how far we are from the production of the new drug against a  parasite like Leishmania based on a  AMP. In different sections the activity of AMPs against Candida species is reported, but Candida is a fungus and not a bacteria and, therefore, it should not be included considering the title.

Some specific aspects to be changed:

line 97 Concerning their mechanisms of action, : this sentence does not make sense, something is missed;

lane 205 Drosophila, …): these dots should be removed;

lane 208 with about 4,000 g/mol molecular weight: this sentence does not make sense;

lane 341 human beings and other organisms [132], animals, plants: why other organisms are differentiated from animals?

lane 507: antimicrobial efficacy against bacteria, Candida albicans: Candida albicans is not a bacteria.

Author Response

Thank you.

Reviewer 2 Report

This manuscript entitled "Activity of antimicrobial peptides (AMPs) against Leishmania, other parasites and bacteria: an overview" delivers interesting review on the control of Leishmania, parasites and bacteria by AMP. 

This manuscript can be accepted to Biomolecules based on the excellence of the manuscript contents, however, before the final decision, it can be improved by adding more information as follows;

  1. Please prepare the information of commercial products controlling Leishmania by AMPs, probably as section 3.
  2. Please prepare the conclusions and future perspectives as section 4.

Author Response

Thank you.

Round 2

Reviewer 1 Report

The paper has been improved from the first version. In the final part of the abstract you should write "against other parasites" and not "against other microorganisms".